# Different Traits, Different Evolutionary Pathways: Insights from *Salamandrina* (Amphibia, Caudata)

**DOI:** 10.3390/ani12233326

**Published:** 2022-11-28

**Authors:** Claudio Angelini, Francesca Antonucci, Jacopo Aguzzi, Corrado Costa

**Affiliations:** 1Salamandrina Sezzese Search Society, Via G. Marconi 30, 04018 Sezze, Italy; 2Consiglio per la ricerca in Agricoltura e l’Analisi dell’Economia Agraria (CREA), Centro di Ricerca Ingegneria e Trasformazioni Agroalimentari, Via della Pascolare 16, 00015 Monterotondo, Italy; 3Instituto de Ciencias del Mar (ICM-CSIC), Paseo Marítimo de la Barceloneta, 37–49, 08003 Barcelona, Spain

**Keywords:** phenetic characters, multivariate clustering, image analysis, local adaptation, evolutionary processes

## Abstract

**Simple Summary:**

In this paper we studied the body and colour features of the endemic amphibian *Salamandrina* across its range. Using a multimodel approach and machine learning clustering, we found much difference relative to body size and shape, as well as colour, among populations and between the habitats in which the individuals dwell, while differences between the two species of *Salamandrina* genus are lower. We suppose that selective pressures are more similar across species’ ranges than within them. Taking into account also previous knowledge on this genus, here we point out that different traits of an organism could result from different evolutionary routes, something not unexpected but often neglected.

**Abstract:**

Species delimitation is often based on a single or very few genetic or phenetic traits, something which leads to misinterpretations and often does not provide information about evolutionary processes. Here, we investigated the diversity pattern of multiple phenetic traits of the two extant species of *Salamandrina*, a genus split only after molecular traits had been studied but the two species of which are phenetically very similar. The phenetic traits we studied are size, external body shape and head colour pattern, in a model comparison framework using non-linear mixed models and unsupervised and supervised clustering. Overall, we found high levels of intra-specific variability for body size and shape, depending on population belonging and habitat, while differences between species were generally lower. The habitat the salamanders dwell in also seems important for colour pattern. Basing on our findings, from the methodological point of view, we suggest (i) to take into account the variability at population level when testing for higher level variability, and (ii) a semi-supervised learning approach to high dimensional data. We also showed that different phenotypic traits of the same organism could result from different evolutionary routes. Local adaptation is likely responsible for body size and shape variability, with selective pressures more similar across species than within them. Head colour pattern also depends on habitat, differently from ventral colour pattern (not studied in this paper) which likely evolved under genetic drift.

## 1. Introduction

Whichever species concept is used, the taxonomic assignment of individuals to a species relies on the (combined) analysis of phenotypic and genetic characters [1,2]. Thanks to spectacular technological advancements in molecular biology, genetics has become the backbone of evolutionary studies more and more during the previous decades. However, researchers have become aware that “gene trees and species trees are not the same” [3], and multilocus approaches are advisable. Furthermore, genetic surveys require well-equipped and funding-sustained laboratories, a combination of conditions not always available. In practice, the phenotype is largely the way by which species are still recognised, at least as the first step, and genetic studies are usually carried out for deeper and further investigation. Therefore, most field ecologists primarily rely on the phenetic species concept.

On the other hand, genetic evidence of speciation stimulates the study of differences among species, and differences or resemblances in turn stimulate evolutionary hypotheses, i.e., speciation pathways. Usually, searching for differences among species leads one to think in terms of “there is no difference” (null hypothesis, *H*_0_) or “there is a difference” (a single alternative hypothesis, *H*_1_), which is null hypothesis testing. However, while the difference (or resemblance) detected by hypothesis testing can be true, it is not necessarily the stronger signal the dataset bears, it could even be the weaker. In fact, any dataset brings several signals, and while it is very difficult to detect the true information (i.e., the strongest signal), it is possible to search for and likely find various signals (i.e., patterns). On the other hand, not all signals have the same strength. Instead, multimodel inference and model selection [4] allow to compare the strength of different signals, and thus each of multiple hypotheses (*H_i_*). Thus, model selection is a natural companion for finding discrete clusters within an apparently homogenous group of organisms or for comparing the partitions within a group based on different biological rationales (species, populations, sexes, ecotypes, etc.).

In this paper, multimodel inference was applied to investigate if any phenetic trait differs between the two only extant species of the genus *Salamandrina*, the Northern Spectacled Salamander *S. perspicillata* (Savi, 1821), occurring from Liguria to northern Campania, and the Southern Spectacled *S. terdigitata* (Bonnaterre, 1789), from Campania to Calabria, in the Italian peninsula. The species are clearly separated taxa based on allozymes [5] and mitochondrial DNA [6,7]. However, nuclear DNA reveals incomplete lineage sorting and a hybrid zone [7,8,9]. The split between the two species happened between 11 and 2.2 mya [6,8], likely as the result of a climatic cooling trend and geographical segregation [8]. It is known that differences in colouration and morphology allow for a percentage of correct species assignment larger than 90% when using discriminant statistics [10,11]. Furthermore, it is known that salamanders’ body size is affected by the type of breeding site, and it shows high intra-specific variability, at least for *S. perspicillata* [12,13].

The main goal of this study is to analyse multiple phenetic traits to evaluate if differences between the two species exist, and if differences due to the habitat are stronger than differences between species. In doing this, we also took into account the contribution to phenetic variability attributable to intra-specific differences at a population level, a factor often underestimated or even neglected when investigating differences between species, which instead turned out to be important.

## 2. Materials and Methods

Three populations of the Northern spectacled salamander *S. perspicillata* and two populations of the Southern spectacled salamander *S. terdigitata* were compared (Table 1). All populations inhabited mixed deciduous forests. The northern species populations were sampled in the Lepini mountains (about 800 km^2^, Lazio region) as this area harbours the largest body size individuals and populations discovered in this species [12,13]. By doing so, we maximized body size differences between species, a caution which allowed us to evaluate the role of species belonging versus habitat and population belonging in shaping salamanders’ bodies (in size). We are aware the study design is incomplete, because a southern pond population is missing (as a matter of fact, very few *terdigitata* populations oviposit in lentic water), however, this is not a limitation for the study, as will result from our findings. Data for both species were collected during oviposition seasons. Only data collected from females were analysed.

### 2.1. Biometries

Animals were captured on sight. We measured the following body features: AG: distance between axilla and groin, CL: cloaca length, HL: head length, HW: head width, No: distance between nostrils, RU: radius-ulna (forearm) length, SVL: snout-vent length, TF: tibia-fibula (leg) length, TL: total length, tL: tail length and tW: tail width (Figure 1, Appendix A). They were measured directly on the individuals after anaesthesia (by using a 0.01% solution of MS-222-Sandoz) with a calliper (to the nearest 0.1 mm), but HL, HW and No were measured from photographs using the software tpsDig2 [14]. SVL and TL showed high collinearity with AG (r_Pearson_ = 0.94 and 0.9 respectively, *p* < 0.001 in both cases) and thus were not used for the main analyses. All salamanders were released at their site after they recovered from the anaesthesia.

We investigated how biometric variation was related to species (*S. perspicillata* or *S. terdigitata*), type of oviposition site (brooks versus lentic water) and population belonging, by using model selection based on an information theoretic approach [15]. Namely, linear mixed-effects models were fitted to evaluate how the three factors explain size (AG) and multivariate morphological variation. Both “species” and “site typology” predictors were used as fixed factors separately and never crossed each other; “population” was always used as a random factor, either nested within “species” and “site typology” or alone. For AG we compared five models (Table 2) (since SVL and TL are often used as proxies for body size, we also show the results of the same analysis for these features in Appendix A). For the analysis of the multivariate morphological variation, we considered all (eight) body features, except AG, and then compared five models built as described above, plus the same five models also including AG as a covariate, as well as a further model with only AG as a continuous predictor, thus obtaining 11 candidate models in total (Table 3). Model comparisons were done using Akaike’s information criterion for small sample size (AICc) and AICc weight, the likelihood that a given model is actually the best among the candidate ones [15]. Analyses were performed in the R environment [16], using the packages lme4 (version 1.1-31, [17]) for fitting the models: lmerTest (version 3.1-3, [18]) was used to obtain the (multivariate) analysis of variance [(M)ANOVA] output from the linear mixed model fitted with lme4, and MuMIn (version 1.9.13, [19]) for the computation of model weight.

### 2.2. Head Colour Topographical Analysis

*Salamandrina* is characterized by a yellow V-shaped patch on the head. Analyses were based on digital photographs (1200 × 1600 pixels; 72 dpi) of the dorsal view of salamander heads. Animals were placed horizontally, as much as possible following the main body axis, the camera was parallel to the animals and a metric reference was placed aside. Then, a geometric morphometric analysis was performed on each picture [20,21,22]. Nine landmarks and 24 equidistant points along the outline (semi-landmarks) [21] from the head of each salamander were digitized using the software TpsDig2 (Ver. 2.17 [14]). All landmark and semi-landmark configurations for each specimen were aligned, translated, rotated and scaled to a unit centroid size by the Generalized Procrustes Analysis (GPA [23]), using the consensus configuration of all specimens as the starting form. Residuals from the fitting were modelled with the thin-plate spline interpolating function [21,22,23]. The shape and head colour pattern of each individual were morphologically adjusted to a standard view by means of the consensus configuration. Before the morphometric standardization, the head colour pattern was manually segmented into pure black and white. The final grey-scale images were 2288 × 1712 pixels at 96 ppi, for an amount of 26,748 pixels constituting the head’s area.

We used the information on position and colour (black or white) of every single pixel of each individual to cluster salamanders on the basis of their head colour pattern by applying both unsupervised and supervised clustering. For unsupervised clustering we compared all the models with a candidate number of clusters from two to six. Candidate supervised clustering was based on: partition from the best model from unsupervised clustering (two clusters), species (two clusters), population belonging (five clusters) and habitat (two clusters). This has been done in the R environment [16] by using the package MixAll (version 1.5.1 [24]). MixAll allows for model-based clustering and classification. Notably, we used multivariate categorical mixture models. For model selection we used the Integrated Completed Likelihood (ICL) criterion [25], a model selection criterion specifically introduced for model-based clustering. Since MixAll does not allow for reducing dimensionality and does not show the main features underlining differences, we applied the Partial Least Square Discriminant Analysis for this purpose. Details and results from this analysis are reported in the Appendix A.

## 3. Results

### 3.1. Biometries

Model selection shows that the effect of both the type of oviposition site and species on salamanders’ body sizes (AG) appear evident only when accounting for intra-specific differences at the population level (Table 2). Based on models’ weight, population belonging is the main determinant for body size, then it depended on site typology (salamanders from brooks are smaller than ones from lentic waters; Appendix A) and finally, there is poorly if any effect from salamander species (second best model, *S. perspicillata* are larger; Appendix A). Nevertheless, the fixed factors always obtained significant *p*-values (marginally so in one case). We want also to stress that our findings are not invalidated by the absence of a *S. terdigitata* pond population from the dataset: since salamanders from ponds tended to be larger, a *terdigitata* pond population would have further made the size of *S*. *terdigitata* similar to *perspicillata*.

Model selection for multivariate morphological variation clearly indicated that it depended on intra-specific population variability and body size (all body features are positively correlated with AG, with r_P_ ranging from 0.37 to 0.84, *p* < 0.001), and the role of both species and oviposition site is poor and only detectable when accounting for body size and population variability (Table 3). At both the univariate (size) and multivariate (morphology) level, species is the predictor that performed the worst. However, fixed factors again achieved significant *p*-values, except in one case.

### 3.2. Head Colour Topographical Analysis

Unsupervised clustering indicates that individuals group in two clusters (ICL = 2,382,622), since the ICL from three clusters (2,393,146) onward increasingly augment. There were associations between these two clusters and habitat (chi-square *p* < 0.01) and population (*p* < 0.001), but not with species (*p* = 0.28). Supervised clustering ranked as best the partition obtained from unsupervised clustering (ICL = 5,441,850), then the partition based on site typology (Table 4).

## 4. Discussion

Both kinds of analyses show that, even though *Salamandrina* species differ, interspecific difference is not the stronger signal the data bear. A one-way (M)ANOVA approach would have detected a significant effect of the fixed factors species and site typology on morphometry for (almost) all alternative models (Table 2 and Table 3), which is hardly useful. On the other hand, testing only one hypothesis (e.g., difference between species) would have led us to neglect other, more likely, hypothesis (e.g., difference between site typology taking into account population, see Table 2). Model selection allowed us to figure out: (i) the importance of population belonging for both univariate and multivariate variability; (ii) the partial importance of the site typology for body size variation (when accounting for population belonging); (iii) the dependence of multivariate variability on population belonging and on body size (two factors that being related reinforce their signal in explaining the multivariate information).

The unsupervised analysis of the head colour pattern found two clusters. These clusters are not randomly associated with site typology and population, but reveal a further, more important, even though not interpretable, pattern. Apart from these clusters, supervised clustering ranked as the best the partition between site typologies, showing that local factors are determinants also for the shape of head spots. Despite the very high difference among the information criterion values, the accuracies of the classifications are rather good for site typology, species and population (taking into account that the probability of random assignment among populations is 20%). Again, also for the head colour pattern, as for body shape and size, testing only one model could have led us to a spurious conclusion, as well as testing more models without an estimator of their relative quality would have been uninformative (see Appendix A). However, it depends on the objective of the investigation: if it is finding a way to distinguish groups, supervised learning based on only one hypothesis can be useful. For example, when applied to *Salamandrina*, it actually allows for a percentage of correct species assignment larger than 90% [10,11]. However, if the issue is to find the most relevant biological pattern, model selection among unsupervised learning and/or supervised models should be the choice.

The problem of statistically approaching unknown classification patterns is emerging with the evolution of machine learning approaches; consequently, clear protocols on whether using supervised or unsupervised techniques are not yet clearly established [26]. Unsupervised learning methods significantly outperform the supervised ones when the number of clusters [27] or the observation attributions [28] to classes are unknown or unclear. On the other hand, supervised methods could produce a more informative output, e.g., finding out the latent dimensions responsible for a given clustering or identifying the main features underlining differences. The combination of supervised and unsupervised approaches is called semi-supervised learning and offers a promising direction of future research [26,29]. We suggest as an optimal workflow, especially when dealing with high multidimensional data, such as data from image analysis: first to use unsupervised modelling, then compare hypotheses by supervising learning and finally, apply to the best clustering a statistical tool which reduces dimensionality and shows the features responsible for differences among clusters. As an example, in the Appendix A we show the implementation of Partial Least Square Discriminant Analysis on our models.

Apart from methodological hints, our analyses yielded also some biological findings. It was already known that *S. terdigitata* is smaller than the northern species [10] as well as that the typology of the oviposition site is even more important in determining the body size [13]. We add to this information the key role played by population belonging in determining both size and shape of salamanders. Despite the fact we opportunistically introduced a bias in the analysis by sampling the largest body size populations of *S. perspicillata*, we found a high level of biometric and morphometric diversity within both *Salamandrina* species compared to diversity between them. This confirms that the selective pressures acting on these traits are more similar across ranges than at a local scale. The split between the two species happened in an allopatric speciation scenario [8], which is expected to cause slow diversification [30,31,32]. In fact, if the physical barrier does not produce any ecological difference between the new-species’ ranges, the species continue to share the same kind of environment, they will still be under the same selective pressures, and traits under such pressures will differ little between species. In this framework, local pressures could become the main ones. Since the *perspicillata* populations we studied are closely related (same mitochondrial haplogroup and microsatellite cluster [8]) and are no more than 4 km away from each other, we are prone to attribute population differences to very local factors (i.e., excluding climate). Based on our results, we suppose that local topography is a candidate to take into consideration. This is in accordance with what was previously suggested, that salamanders from brook populations experience a lower survival rate due to floods, thus leading to a smaller body size [12]. However, lower survival could cause smaller body size just as a demographic by-product (younger individuals are smaller) and/or by adaptative reaction leading to earlier sexual maturity [33] and thus smaller adult size ([34] and reference therein). It is known that *Salamandrina* shows high phenotypic plasticity, with close populations having different oviposition periods [35,36], however, only knowledge on individual ages can disentangle the role of demography and reaction norms on body size and shape variability, as well as the validation of our hypothesis requires studies involving (local) environmental factors.

The V-shaped patch on the head depends mostly on the type of oviposition site, little if any on species and even less on population. It is difficult to explain this finding, because no hypothesis exists about the role of the yellow head patch (we could just speculate it is a disruptive or distractive colouration). Differently than the head yellow patch, the extension of red colouration on the tail and the ventral colour pattern proved to efficiently discriminate between *Salamandrina* species [10,11]. The discriminatory performances of such colour features suggest that they are not subjected to the same selective pressures as the morphometric and colouration traits we analysed in the current study. Likely, the anterior part of the ventral colouration is involved in intra-specific communication [11,37,38,39], as well as the tail [40]. The speciation scenarios suggested for *Salamandrina* [8,9] involve the reduction of population size in separated refugial areas, from which both species colonized the extant ranges. In particular, both studies pointed out that the *perspicillata* lineage survived in a single refugial area. In the small, ancestral population, the importance of genetic drift was likely higher than natural selection. For any colour pattern involved in intraspecific communication, the random process would have been positively reinforced by the efficacy of signalling [41], thus leading to fixation of colour differences between the two *Salamandrina* species. Something that did not happen for the head V-patch, or that has been blurred later by any selective pressures.

## 5. Conclusions

Our findings confirm that within-species variability is not negligible when studying phenetic and functional differences between species, and that such variability can even challenge the discrimination between species [42,43]; thus, taking into account differentiation at population level is an important caution. We also showed how different phenotypic traits of the same organism could follow different evolutionary routes. On one hand, genetic drift probably led to strong differentiation of some, but not all, colour patterns between *Salamandrina* species. On the other hand, natural selection seems to act on size and shape of individuals in similar ways across the genus’ range, but depending on local factors, it produces ecotypes that differ at least as much as the species and are shared between them. Focussing the analysis only on species discrimination would have hidden more important differences within a genus, as well as intriguing insights.

## Figures and Tables

**Figure 1 animals-12-03326-f001:**
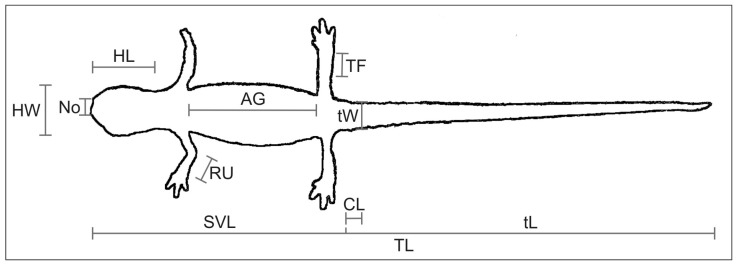
Biometric features of *Salamandrina perspicillata* and *S. terdigitata* used for the study: AG: distance between axilla and groin, CL: cloaca length, HL: head length, HW: head width, No: distance between nostrils, RU: radius-ulna (forearm) length, SVL: snout-vent length, TF: tibia-fibula (leg) length, TL: total length, tL: tail length, tW: tail width.

**Table 1 animals-12-03326-t001:** Summary information about the sites of *Salamandrina perspicillata* and *S. terdigitata* used in this study; site codes are the same as in Appendix A.

Species	Site (Code)	Location	Habitat Features	Sampling Year and Size
*S. perspicillata*	Acqua della chiesa (AC)	Lepini Mountains, Latium	Trough, 900 m a.s.l.	2007 *n* = 28
*S. perspicillata*	Ciccopano nuove (CN)	Lepini Mountains, Latium	Rocky spring ponds, 700 m a.s.l.	2006, 2007 *n* = 19
*S. perspicillata*	Sant’Angelo (SA)	Lepini Mountains, Latium	Brook, 940 m a.s.l.	2007 *n* = 27
*S. terdigitata*	Torrente Cerasuolo (TC)	Picentini Mountains, Campania	Brook, 750 m a.s.l.	2007 *n* = 41
*S. terdigitata*	Torrente Rosa (TR)	Pollino, Calabria	Brook, 550 m a.s.l.	2007 *n* = 41

**Table 2 animals-12-03326-t002:** Model ranking of candidate models used to analyse the dependence of axilla-groin distance of salamanders based on the species (*Salamandrina perspicillata* or *S. terdigitata*), the type of breeding site (brook or lentic water) and their population (five populations). Species and site type were used as fixed factors, population as a random factor (when together, the random factor is nested in the fixed one). AICc is the Akaike’s information criterion for small sample size, w is the model weight. *p*-value refers to the fixed factor for the given ANOVA model.

Model	AICc	W	*p*-Value
site typology:population	609.25	0.74	0.018
species:population	612.38	0.16	0.1
population	613.27	0.1	
site typology	644.94	0	<0.001
species	668.81	0	<0.001

**Table 3 animals-12-03326-t003:** Model ranking of candidate models used to analyse the multivariate dependence of body features (see text for details) of salamanders based on the species (*Salamandrina perspicillata* or *S. terdigitata*), the type of breeding site (brook or lentic water), their population (five populations) and the individual axilla-groin distance (AG; “*” denote the use of AG as covariate). Species, site type and axilla-groin distance were been used as fixed factors, population as random factor (when together, the random factor is nested in the fixed one). AICc is the Akaike’s information criterion for the small sample size, w is the model weight. *p*-value refers to the categorical fixed factor species or site typology for the given MAN(C)OVA model.

Model	AIC	W	*p*-Values
population*AG	4260.98	0.986	
species*AG:population	4270.87	0.008	=0.04
site typology*AG:population	4271.35	0.006	=0.04
species*AG	4448.34	0	<0.001
site typology*AG	4470.28	0	<0.001
AG	4722.57	0	
population	4851.38	0	
site typology:population	4862.03	0	=0.03
species:population	4863.34	0	=0.14
site typology	5509.29	0	<0.001
species	5718.67	0	<0.001

**Table 4 animals-12-03326-t004:** Model ranking of candidate models for supervised clustering. Unsupervised partition refers to the best clustering obtained from unsupervised clustering. Accuracy is the classification accuracy of the model.

Model	ΔICL	Accuracy
unsupervised partition	0	0.97
site typology	32,919	0.72
species	44,959	0.73
population	116,985	0.52

## Data Availability

The data presented in this study are partly available in the Appendix A, and are available on request from the corresponding author.

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
