# Peer review of "Different Traits, Different Evolutionary Pathways: Insights from Salamandrina (Amphibia, Caudata)"

_animals, 2022, doi:10.3390/ani12233326_

Round 1

Reviewer 1 Report

I don’t find the manuscript attractive to the readers, however the paper is well written. In the introduction it should be explained why there is still interest in splitting the genus using morphology if there is genetic support. 

Data analysis. I don’t understand why the authors used MANOVA after model selection? 

I don’t know this unsupervised and supervised clustering. Please provide some references. Line 161 which models? The same on which you did model selection? 

Line 177 show the mean (SD) and the significance test to support the statement 

Table 2. It is confusing because authors combined in the same tabele the model selection results and the ANOVA result of a particular model. I would present the results separated. Same for Tabele 3

Author Response

Reviewer 1 (replies are preceded by bold *R#, the numbering is the same as the cover letter)

- I don’t find the manuscript attractive to the readers, however the paper is well written. In the introduction it should be explained why there is still interest in splitting the genus using morphology if there is genetic support.

*R17: We had no interest in splitting the genus using morphology. In the Introduction we explained that it is interesting to evaluate if actually phenetic differences between the two species exist according to genetic splitting and if phenetic differences due to the habitat are stronger than difference due to genetic.

- Data analysis. I don’t understand why the authors used MANOVA after model selection?

*R18: our study is in part methodological, the answer to this question can be found in the first paragraph of the Discussion.

- I don’t know this unsupervised and supervised clustering. Please provide some references. Line 161 which models? The same on which you did model selection?

*R19: i) Sorry, I am not sure I understand the first part of the comment. If the reviewer refer to the algorithm used by MixAll, it is already quoted as [24] in the text. ii) Yes, the best model from unsupervised clustering (as now is specified in new lines 169-170).

- Line 177 show the mean (SD) and the significance test to support the statement

*R20: mean±SE is reported in supplementary Table S1 (information now added to the text, see new lines 185 and 187), significances are in table 2.

- Table 2. It is confusing because authors combined in the same tabele the model selection results and the ANOVA result of a particular model. I would present the results separated. Same for Tabele 3

R21*: the manuscript is actually based on model selection, (M)ANOVA results are reported only for comparison (please, see also *R18 above); when appropriate, results are reported in the text (see *R20 just above). Thus, I think that splitting the tables is not useful for the manuscript.

Reviewer 2 Report

The paper investigates the possible causes of phenetic variation between the two species of Salamandrina by morphometric techniques. The problem is well addressed and the statistic framework is adequate, supporting authors' conclusions. Anyway, I've found some minor problems:

1) Both species are protected by Italian laws: permissions are required for collecting, manipulating and anesthetizing them. As this Journal's Ethical Guidelines require, authors should indicate in the text that their research obtained approval by local authorities and give reference of the permission documents.

2) In Table 1, it is unclear what the letters in brackets after the sampling localities mean, they do not seem to be abbreviations for Provinces.

My overall recommendation is: minor revisions required.

Author Response

Reviewer 2 (replies are preceded by bold *R#, the numbering is the same as the cover letter)

The paper investigates the possible causes of phenetic variation between the two species of Salamandrina by morphometric techniques. The problem is well addressed and the statistic framework is adequate, supporting authors' conclusions. Anyway, I've found some minor problems:

1) Both species are protected by Italian laws: permissions are required for collecting, manipulating and anesthetizing them. As this Journal's Ethical Guidelines require, authors should indicate in the text that their research obtained approval by local authorities and give reference of the permission documents.

*R22: yes, we forgot to include the permission in the manuscript. It was required soon by the Editor and now it is also in the manuscript (new lines 342-343).

2) In Table 1, it is unclear what the letters in brackets after the sampling localities mean, they do not seem to be abbreviations for Provinces.

*R23: I added a sentence in the table caption which should better explain the code (new line 108).